

# High-resolution typhoon precipitation integrations using satellite infrared observations and multi-source data

You Zhao[1,2], Chao Liu[1,2], Di Di[1,2], Ziqiang Ma[3], Shihao Tang[4]

[1]School of Atmospheric Physics, Nanjing University of Information Science and Technology, Nanjing 210044, China.
[2]Key Laboratory for Aerosol-Cloud-Precipitation of China Meteorological Administration, School of Atmospheric Physics, Nanjing University of Information Science and Technology, Nanjing 210044, China.
[3]Institute of Remote Sensing and Geographical Information Systems, School of Earth and Space Sciences, Peking University, Beijing, 100871, China.
[4]Key Laboratory of Radiometric Calibration and Validation for Environmental Satellites, National Satellite Meteorological Center, China Meteorological Administration, Beijing 10008, China.

*Correspondence to*: Chao Liu (chao_liu@nuist.edu.cn)

**Abstract** Typhoon-related precipitation over land can result in severe disasters such as floods and landslides, and satellites are a valuable tool to estimate surface precipitation with high spatial-temporal resolutions. This study develops a high-
resolution surface precipitation integration framework to combine observations from geostationary Fengyun-4A/ Himawari-8 (F4/H8) satellite radiometers, high-density rain-gauge observations (or Integrated Multisatellite Retrievals for GPM (IMERG) data) and atmospheric reanalysis (ERA5) based on a random forest (RF) algorithm. The RF methods fuse cloud and atmospheric features from radiometric observations and reanalysis information, and the intensity and spatial distribution of rainfall can be revealed by high-density rain-gauge or IMERGE observations. We take three typhoons made landfall in
South China in 2018 as examples. Both F4- and H8-based results using rain-gauge data as the predictand show excellent results with correlation coefficients (R) with the references ~0.75 and probabilities of detection (POD) as large as 0.98, higher than current satellite-only results. However, if the IMERG data are used as the predictand, the corresponding R and POD drop to ~0.5 and 0.93, respectively, due to the uncertainties related to IMERG retrievals. By carefully choosing the predictors, the RF algorithm can clearly summarize the information from satellite observations, surface observations and
atmospheric reanalysis, resulting in precipitation results that are highly consistent with the actual ground observations. Our proposed integration framework can not only reconstruct hourly surface precipitation estimation datasets at high-spatial resolution for historical typhoon studies but also potentially monitor the fine- resolution surface precipitation of frequent typhoons in near-real time.

## 1 Introduction

Typhoon, also referred to as tropical cyclones, is a high-impact atmospheric phenomena, and cause some of the most significant socioeconomic damage due to their intense winds, immense storm surges and flood-inducing rainfall [Wang er al.,



2019; Rappaport et al., 2014; Negri et al., 2005]. There are over eighty typhoons every year globally, and approximately one-third of which is generated in the Northwest Pacific (NWPAC) [Chan et al., 2005]. Furthermore, although the number of typhoons in the NWPAC is exhibiting a decreasing trend with global warming, their average intensity is increasing, resulting

in even more severe damage in coastal areas [Webster et al., 2005; Emanuel et al., 2013; Kang et al., 2016; Cai et al., 2004; Ho et al., 2021]. Unfortunately, because of the high spatial-temporal variability and complex physical processes, rainfall still accounts for one of the largest uncertainties in the forecasting of tropical cyclones [Su et al., 2012; Tu et al., 2013], and its high-quality observations are also limited. Thus, precipitation observations with a high spatial-temporal resolution during typhoon periods play an important role in not only researches on typhoon precipitation characteristics but also in disaster

prevention.

Ground rain-gauges, weather radars and satellites are commonly used to acquire precipitation measurements [Villarini et al., 2008], and they all have their unique advantages and disadvantages. As the most direct and accurate method for measuring the amount of rainfall to reach the ground, rain-gauges cannot capture the complete spatial distribution of rainfall due to the limited number of stations, especially in oceanic, mountainous and polar regions [Susana et al., 2013; Gires et al., 2012;

Rodriguez et al., 2019; Looper et al., 2012]. In contrast, microwave radars can survey larger areas and can better capture the spatial variability of rainfall fields; however, the accuracy of radar-based measurements is highly influenced by electromagnetic attenuation and the uncertainty in the relationship between the radar reflectivity factor and precipitation, particularly under extreme rainfall conditions [Marra et al., 2015; Bárdossy et al., 2017]. Satellite-based quantitative precipitation estimation (QPE) can be implemented on a large scale with a high spatial-temporal resolution, offering large

scale capability with high spatial-temporal resolutions [Tang et al., 2016; Wang et al., 2018; Jozaghi et al., 2019], but quantitatively inferring the amount of surface precipitation from space is still a serious challenge, especially during typhoon periods. Nevertheless, with the continuous improvement of meteorological satellites, satellite-based QPE technologies have undergone considerable development [Boushaki et al., 2009; Nguyen et al., 2018]. Various models have been developed to generate satellite-based QPE products by relying on the relationships between passive infrared/microwave observations and

precipitation; typical products include the Tropical Rainfall Measuring Mission Multisatellite Precipitation Analysis (TRMM TMA) [Huffman et al., 2007; Liu et al., 2016], Integrated Multisatellite Retrievals for the Global Precipitation Measurement (GPM) Mission (IMERG) [Huffman et al., 2015; Wang et al., 2017], Climate Prediction Center morphing technique (CMORPH) [Joyce et al., 2004] and Global Satellite Mapping of Precipitation (GsMaP) [Aonashi et al., 2009; Ushio et al., 2009]. At present, machine learning (ML) methods are widely used to establish the relationship between rainfall rates and

satellite passive spectral observations as well [Hong et al., 2004; Kühnlein et al., 2014; Sehad et al., 2017; Ahmed et al., 2020; Mountrakis et al., 2011; Albawi et al., 2017; Kühnlein et al., 2014]. Min et al. used a random forest (RF) algorithm to establish the relationship between spectral imager observations and numerical weather prediction results, and quite reasonable rainfall area and intensity can be obtained [Min et al., 2018]. Wang et al. used convolutional neural network (CNN) to establish a high-quality precipitation dataset based on Chinese FengYun geostationary satellite [Wang et al., 2017].

Most those studies intended to develop a general model for precipitations of all kinds globally or over large regions, while



the different responses of precipitation caused by different kinds of cloudy and atmospheric conditions may limit their accuracy.

Meanwhile, typhoons not only result in heavy and wide range precipitations, but also thick clouds that significantly influence/blocks satellites for surface variable observations. Considering the complex and unclear relationship between precipitation and clouds, the QPE product particularly designed for typhoon precipitation remains limited, leaving a gap in accurate and high resolution surface precipitation. Meanwhile, FengYun-4A (F4) and Himawari-8 (H8) are new-generation geosynchronous equatorial orbit (GEO) satellites launched by the Chinese and Japanese meteorological agencies, respectively [Min et al., 2017; Bessho et al., 2016]. These two satellites cover a similar observational region of the NWPAC and a large portion of East Asia, and their high temporal and spatial resolutions are favorable for continuously monitoring of NWPAC typhoons [Ma et al., 2021; Honda et al., 2018]. Thus, this study employs a ML technique to integrate F4/H8 radiometer observations and multi-source datasets to develop better surface precipitation integration algorithms particularly for typhoon precipitation, and, more importantly, investigates how multi-source (ground-based, satellite and reanalysis) data improve and contribute the surface precipitation integration differently.

## 2 Data and Study area

### 2 Data

To better estimate surface precipitation during the typhoon period, we would exert the advantages of multi-source data for different atmospheric variables, while also include only popular and public-available ones for the generality of the method. Thus, besides the F4/H8 radiometer measurements, we would consider ground-based observations, atmospheric reanalysis and existing satellite-based surface precipitation estimations.

F4 and H8 can observe most of NWPAC and East Asia, while, over the region, South China is frequently influenced by the landfall of typhoons, resulting in serious disasters and casualties. The spectral radiometer onboard F4, namely the Advanced Geosynchronous Radiation Imager (AGRI), has six visible and near-infrared channels and eight thermal infrared channels. The Advanced Himawari Imager (AHI) onboard H8 has a similar channel design, and is equipped with two additional thermal infrared channels. To develop an all-day available integration algorithm, we consider only the AGRI and AHI infrared channel radiances and their combinations. For a fair comparison between the F4/AGRI- and H8/AHI-based algorithms, only the channels that are similarly equipped on both instruments are used, including the two water vapor channels (WV1 and WV2) and four longwave infrared channels (LW1-LW4). The central wavelengths of each channel considered in our integration algorithm are shown in Table 1, and it becomes interesting whether such small differences on the channel characteristics as well as those on channel spectral response functions and geolocation would influence the surface precipitation estimation.

Passive spectral observations provide mostly the spatial distribution of clouds as well as cloud top properties, whereas surface precipitation is highly related to the atmospheric conditions. Thus, our algorithm also adopts atmospheric variables



such as the column water vapor content, cloud water content, atmospheric profiles and so on. These variables are obtained from the state-of-the-art atmospheric reanalysis data, i.e., the European Center for Medium-Range Weather Forecasts
(ECMWF) Reanalysis v5 (ERA5), the latest atmospheric reanalysis product developed by ECMWF [Hersbach et al., 2020]. We consider nine variables from ERA5 related to typhoon precipitation to infer the general atmospheric conditions within the integration region [Min et al., 2018]. For example, the convective available potential energy (CAPE) is an indication of the instability (or stability) of the atmosphere and can be used to assess the potential for the development of convection, which can lead to heavy rainfall, thunderstorms and other severe weather. The K-index, calculated from the temperature and
dew point temperature in the lower part of the atmosphere, is a measure of the potential for a thunderstorm to develop. The total column rain water (TCRW) is the total amount of water in droplets of raindrop size (which can fall to the surface as precipitation) in a column extending from the surface of the Earth to the top of the atmosphere. Total precipitation (TP) is the accumulated liquid and frozen water (comprising both rain and snow) that falls to the Earth's surface. The total column liquid water (TCLW) is the total amount of supercooled water in a column extending from the surface of the Earth to the top of the
atmosphere. Furthermore, four basic atmospheric variables are considered, namely, the relative humidity at 850 hPa (R850) and 950 hPa (R950) and the temperature at 850 hPa (T850) and 950 hPa (T950), which describe the humidity and temperature states in the lower atmosphere.

## 2.2 Study area

Accurate precipitation observations are needed as the references to train ML-based models, and thus are essential for high-
quality integration. Two kinds of data are considered: high-density ground rain-gauge data and IMERG estimates. Offering some of the most reliable and fundamental precipitation observations, rain-gauge data are obtained from the National Meteorological Information Center of the China Metrological Administration (CMA). Due to the high frequency of rainfall and dense urban distribution in South China, automatic rain-gauge stations are densely distributed throughout the region. Within the land area of the studied region ($79\times10^4$ km$^2$), there are a total of 5024 rain-gauge stations, the distribution of
which is shown in Figure 1. However, such a high density of ground-based observations may not always be feasible, so another type of data that is more commonly used in ML-based precipitation estimation is also considered, i.e., the IMERG final run-calibrated precipitation data. IMERG provide gridded precipitation estimation from PMW sensors on board various satellites in the GPM constellation and IR-based observations from geosynchronous satellites [Liu et al., 2016, Tang et al., 2016], and is used as one of the most reliable precipitation datasets. IMERG has a half-hourly temporal interval with a
maximum rain rate of 50 mm/h and cover the area between the latitudes of 60°S and 60°N [Min et al., 2018].

This study investigates three typhoon events that made landfall in South China in 2018. Information on these three typhoon events is provided in Table 2, and the paths of the three typhoons are illustrated in Figure 1. We consider the area within the latitudes 15°-27°N and the longitudes 105°-125°E, including a large part of South China (15°-27°N; 105°-122.5°E) and NWPAC, which are completely covered by the observation areas of F4 and H8. We consider the whole processes from



landing to dissipation of the three typhoons, model performance for a total of ~120 hours would be considered, resulting approximately 600,000 hourly rain rate observations from the ground rain-gauge stations.

## 3 RF-based typhoon precipitation integration framework

### 3.1 Establishment of surface precipitation integration algorithm

The RF algorithm is an ML method widely used in the inversion of meteorological elements and has been proven to perform

well in applications such as the estimation of precipitation, detection of clouds, and inversion of PM2.5 concentrations [Oscar et al., 2020; Tan et al., 2021; Liu et al., 2021; Guo et al., 2021]. Thus, this study used this simple but promising RF algorithm to establish the nonlinear relationships among surface precipitation, satellite observations and atmospheric characteristics.

In the framework of RF algorithms, two types of data are utilized for the integration model: the predictor and the predictand.

To study the influences of different satellite observations (F4/AGRI vs. H8/AHI) as the predictor and different precipitation data (rain-gauges vs. IMERG) as the predictand on the surface precipitation estimation algorithm, we established four independent surface precipitation integration models. The models will be referred to as F4-based and H8-based ones according to the satellite data used in the predictor, and, similarly, the models using rain-gauge observations and IMERG data as the predictand will be mentioned as RG-based and IM-based ones, respectively. Thus, a total of four models, i.e., F4-

RG, F4-IM, H8-RG and H8-IM, are developed and studied. Table 3 differentiates these four models. It is worth noting that the only difference between the rain-gauge-based models and the IMERG-based models is the predictand used in the models, as will be described in more detail in the following.

The predictors for the surface precipitation integration algorithm include the geographic location, radiometer observations and atmospheric reanalysis data. By testing the performance of the RF-based models with different combinations of

variables, our final models consider a total of 21 variables, as listed in Table 4. The rain-gauge observations and IMERG estimates are used to provide corresponding surface precipitation as model predictand. It should be noted that only F4/AGRI observations are used in F4-RG and F4-IM, and only H8/AHI observations are used in H8-RG and H8-IM. Considering the uncertainties related to IMERG estimated precipitation, we use ground rain-gauge data to validate all four models.

Figure 2 illustrates the general flowchart of our surface precipitation integration algorithm, including the model development

and surface precipitation estimation. During the model development, both ERA5 data and satellite data are collocated with ground high-density rain-gauge data to obtain rain-gauge-based training datasets. During the collocation processes, the satellite data have a high spatial resolution (4 km for F4/AGRI and 5 km for H8/AHI), and the average radiances values of the nine satellite pixels (3×3 pixels) closest to each ground station are collocated to represent the satellite data corresponding to that station. In contrast, ERA5 has a much lower spatial resolution (0.25°), so the atmospheric variables from the ERA5

grid point closest to each ground station are used. In other words, all the aforementioned data are collocated with the ground rain-gauge stations, and the resulting datasets are used for further training and model development. For a fair comparison,



the IMERG precipitation estimates are also collocated with the rain-gauge observations, resulting in a dataset with the same spatial-temporal sampling interval as that of the rain-gauge-based models for model training. Thus, the F4-IM/H8-IM and F4-RG/H8-RG models have completely the same predictor dataset, and differ only with regard to the predictant, i.e.,
replacing rain-gauge observations with the corresponding IMERG estimates for F4-IM/H8-IM.

With the training datasets obtained above, the RF algorithm is used for the training phase, and the four surface precipitation integration models (F4-IM, H8-IM, F4-RG and H8-RG) are developed. There are two important RF parameters that need to be considered: the number of trees to grow (n_estimators) and the number of variables randomly sampled as candidates at each split (max_features). These two parameters affect the computational efficiency of the model, but have little influence on
the prediction results of the RF models. To ensure that the four models are comparable, the RF parameters are fixed to be the same. Considering the accuracy, computational efficiency and comparability of the models, the number of trees to grow (n_estimators=1000) and the number of variables randomly sampled as candidates at each split (max_features=10) are both fixed for the four models. Changes in these two parameters have little influence on our modeling results, so we will not discuss the sensitivities of results to the RF parameters.

The right part of Figure 2 depicts the procedure for the pixel-level component of our surface precipitation integration algorithm. At this stage, linear interpolation is employed to obtain the ERA5 atmospheric variables at each F4/H8 grid point. Then, the surface precipitation integration results from the direct RF-based integration models are classified into two sets: pixels with precipitation and those without precipitation. Considering that the minimum precipitation resolution of a rain-gauge is 0.1 mm/h, pixels with a fused rainfall rate below 0.1 mm/h are defined as those without precipitation, while the
integration results are retained for pixels with an estimated rainfall rate greater than 0.1 mm/h. This threshold of 0.1 mm/h ensures a high probability of distinguishing precipitation from non-precipitation.

To fairly evaluate the performance of our surface precipitation integration products, we consider the rain-gauge observations to be the "truth" and perform 10-fold cross-validation (10-cv) to ensure the independence between the training and testing datasets. In other words, the original training dataset is evenly divided into ten parts, one of which is taken as the testing
dataset each time (without repetition), while the remaining nine are taken as the training dataset; this process is repeated ten times. We adopt four popular parameters to quantify the model performance, including two categorical parameters (the probability of detection (POD) and false alarm ratio (FAR)) and two statistical parameters (the correlation coefficient (R) and root mean square error (RMSE)) between our results and the truth [Ebert et al., 2007; Mecikalski et al., 2008]. POD and FAR both range from 0 to 1, where larger POD and lower FAR values correspond to the better identification of precipitation
events.

**3.2 Test and evaluation of the RF-based integration algorithm**

Considering the distribution of precipitation throughout the year and the amount of damage caused by precipitation, the distribution of CMA ground rain-gauge stations is clearly uneven, with many more situated in eastern China than in western China. Because we employ ground rain-gauge observations for the integration, so their "resolution" is crucial for





representing the spatial distribution of precipitation. To explore the impact of station density on the integration of typhoon precipitation, we select data from different numbers of stations to build training datasets of different sizes. The number of stations considered increases from 100 to 5000 with an interval of 100. Figure 3 shows the influence of the number of stations considered on the model training results. The solid lines represent the rain-gauge-based model results, while the dashed lines represent the IMERG-based results; the blue and red colors denote the F4-based and H8-based results, respectively. The evaluation parameters of all four models exhibit similar trends with an increasing number of stations; i.e., the integration results become more accurate due to both additional input data and a better representation on rainfall spatial distribution. The POD values of the four models are all close to 1, and vary little with the number of stations, while the FAR of the four models decreases with an increase in the number of stations. The two statistical metrics fluctuate greatly when the number of stations is less than ~1000, i.e., $12.6 \times 10^{-4}$/km$^2$. When observations from more than 1000 stations are considered, the R/RMSE gradually increases/decreases with an increasing number of stations. In general, in the estimation of typhoon precipitation, when the number of stations covering the region exceed 1000, rain-gauge observations can generally reflect the spatial distribution of precipitation and thus can be used for surface precipitation integration. Of course, an increase in the number of stations does improve the integration performance, but only to a limited degree. Nevertheless, to ensure optimal model performance, this paper selects the data from all available stations.

The scatter plots in Figures 4a–d quantitatively compare the hourly precipitation estimated by our surface precipitation integration models with the corresponding predictand in the 10-cv testing datasets. The two IMERG-based models (F4-IM and H8-IM) outperform the rain-gauge-based models (F4-RG and H8-RG) in the testing datasets according to the values of the POD (0.98 vs. 0.95, 0.97 vs. 0.95), FAR (0.25 vs. 0.46, 0.25 vs. 0.46), R (0.89 vs. 0.79, 0.89 vs. 0.79) and RMSE (1.25 mm/h vs. 2.19 mm/h, 1.25 mm/h vs. 2.17 mm/h). This may be because the satellite-based precipitation data of IMERG as the predictand may demonstrate better spatial consistency with the satellite observations than the rain-gauge observations. In contrast, due to the similar satellite channels used in the prediction variables, the results based on different satellites are very similar in both the rain-gauge-based and the IMERG-based models. However, whether the integration results of these four models can accurately reflect the real ground precipitation will be discussed in Section 4.

To better understand the surface precipitation integration models, Figure 5 shows the importance of all the predictors considered in the four models. The importance of each variable is given in the form of the mean accuracy decrease (%IncMSE), which can represent the relative contribution of each variable, and the sum of the importance of all variables is 100%. For F4-RG and H8-RG, the most important input variables are TCRW and TP, which represent the potential precipitable water in the atmosphere and the sum of large-scale precipitation and convective precipitation given by the atmospheric reanalysis. Geographical location data are also highly important, perhaps because the obvious spatial patterns of the typhoon precipitation could be captured by the RF-based training. The most important radiometer observations are a few BTs, such as BT12.0-BT10.7 for F4-RG and BT6.25-BT10.7 for H8-RG, both of which are related to the state of water vapor in the atmosphere. It is interesting that the important variables of F4-IM/H8-IM differs significantly from those of F4-RG/H8-RG.



## 4 Application of RF-based surface precipitation integration models during typhoon periods

To evaluate the surface precipitation integration results, Figure 6 shows the ground rain-gauge observations and the results from the four integration models at three time steps (from top to bottom) during Typhoon Mangkhut, which made landfall at 0900 UTC on 16 Sep 2018. Three typical cases before the typhoon made landfall (0100 UTC on 16 Sep 2018), after the typhoon made landfall (1200 UTC on 16 Sep 2018) and as the typhoon dissipate (2300 UTC on 17 Sep 2018) are illustrated. Before the typhoon made landfall, the number of stations with precipitation on land was small, and heavy precipitations (e.g.,

stations with rain rates greater than 20 mm/h) was concentrated mainly over the coastline. After the typhoon made landfall, the precipitation area increased significantly, and both the extent of heavy precipitation area and the rain rates increased, with over 29 stations with recording rain rates exceeding 30 mm/h. Before the typhoon started to dissipate, the center of heavy rainfall disappeared, although there were still large regions of weak precipitation. At all three time steps, the four models consistently yield reasonable typhoon precipitation intensities and spatial distributions, especially the F4-RG and H8-

RG, whereas F4-IM and H8-IM slightly overestimate the precipitation extent and can hardly represent the heavy precipitation centers.

To quantitatively understand the abovementioned differences, Figure 7 illustrates the differences in hourly precipitation between our integration results and ground rain-gauge observations at the same three time steps in Figure 6. Warm colors indicate stations with surface precipitation integration results larger than the rain-gauge observations, while cold colors

indicate those with underestimated surface precipitation integration results. The differences between the F4-RG/H8-RG results and rain-gauge observations are mostly within 2 mm/h, significantly smaller than the differences from the F4-IM/H8-IM results. However, all four models tend to underestimate the precipitation intensity in the area of heavy precipitation, while overestimating the precipitation at the rain-gauge stations with relatively weak precipitation (rain rates below 5 mm/h). Ultimately, more than 70% of the surface precipitation was overestimated because there were far more stations experiencing

weak precipitation than there were recording heavy precipitation within the studied region.

Table 5 summarizes the percentages of the precipitation that was overestimated by the four integration models during the three typhoon precipitation periods. The percentages in this table denote ratios of the numbers of our integration results larger than the true values (i.e., rain-gauge observations) to the total amount of data at the particular rainfall rate. As our precipitation differences are strongly consistent with the rainfall rates, we divided the results into two groups: one group for

stations with rainfall rates smaller than 5 mm/h and the other group for those with rainfall rates larger than 5 mm/h. For rain-gauge precipitation less than 5 mm/h, more than ~85% of these data are overestimated. However, for rain rates greater than 5 mm/h, only approximately 20% of ours results are larger than the true values, which means that the rate of underestimation of our models is approximately about 80%.

Figure 8 compares the daily precipitation of our four models during Typhoon Mangkhut on 16 Sep 2018. The daily

precipitation results of based on the rain-gauge predictand (F4-RG and H8-RG, top panels) agree closely, and similar consistency is noticed between the F4-IM and H8-IM results (bottom panels). However, the rain-gauge-based results differ





substantially from the IMERG-based results. The daily precipitation distributions of F4-RG and H8-RG indicate that during Typhoon Mangkhut, the precipitation over land was concentrated mainly in Guangdong Province, and the daily precipitation in some areas surpassed 200 mm/day. In contrast, F4-IM and H8-IM overestimate most of the daily precipitation on land but

significantly underestimate the daily precipitation over 200 mm/day. It is worth noting that all these four models can give precipitation distribution on land and ocean. Unfortunately, there are almost no rain-gauges on the ocean, and the distribution of rain-gauges has a great influence on the results of precipitation integration results, so the inversion accuracy of precipitation on the ocean needs to be further verified.

The spatial distributions of the above biases in our surface precipitation integration results in daily scale are shown in Figure

9. In general, both the rain-gauge-based models and the IMERG-based models share considerable similarities in their spatial distribution of the bias. For the two rain-gauge-based models, the average errors at most stations are between -20 mm/day and 20 mm/day, while the errors in the two IMERG-based models are significantly larger than those in the two rain-gauge-based models. Moreover, ~75% of the stations in all four models overestimate the rain rate, which is consistent with the analysis of the aforementioned hourly scale results.

For more sample validation, Figure 10 shows the surface precipitation integration results from our integration models against ground rain-gauge observations at both hourly (top panels) and daily (bottom panels) scales for all three landfalling typhoon events. The surface precipitation integration results and rain-gauge observations during all three typhoon events are compared simultaneously, with the color bars indicating the occurrence frequency on a log scale at intervals of 0.5 mm/h on the hourly scale and 5 mm/day on the daily scale. In general, the surface precipitation integration results from the rain-

gauge-based models show better consistency with the rain-gauge observations than those from the IMERG-based models, not only at the hourly scale but also at the daily scale.

To explore the temporal characteristics of the four surface precipitation integration models, Figure 11 illustrates time series plots of the four evaluation metrics at the hourly scale during the three typhoon events. The solid and dotted lines represent the RG- and IMERG-based models, respectively, and the blue and red lines represent the F4- and H8-based models.

Generally, the rain-gauge-based models (F4-RG and H8-RG) perform significantly better than the IMERG-based models (F4-IM and H8-IM) with relatively better classification metrics without apparent fluctuations and with better statistical metrics. Because the rain-gauge-based models use ground rain-gauge observations as the predictor while the IMERG-based models use IMERG estimates based on satellite observations as the predictor, the surface precipitation integration results of these two kinds of models differ greatly. In addition, because of the similar satellite observation channels adopted for the

model development, the F4- and H8-based models yield very similar surface precipitation integration results.

Note that all surface precipitation integration models perform better during Ewiniar and Mangkhut than during Bebinca, with the former two having higher POD and R values and lower FAR values; this was caused by the uneven distribution of surface precipitation. As shown in Table 1, there was either light precipitation or no precipitation over most of the land area during Bebinca, accounting for 78% and 18%, respectively. Therefore, the surface precipitation integration results of all

models for the precipitation process during Bebinca are relatively poor.



## 5 Summary

This paper proposes an RF-based surface precipitation integration framework for typhoons making landfall with geostationary spectral radiometer observations, atmospheric reanalysis data, high-density rain-gauge observations and IMERG estimates. To develop the model, we consider either F4 or H8 observations as the predictor and either rain-gauge

observations or IMERG estimates as the predictand. Thus, the performances of four models, i.e., F4-RG, H8-RG, F4-IM, and H8-IM, are systematically evaluated. All four models are capable of capturing precipitation events caused by typhoons making landfall. Regardless of hourly precipitation or daily precipitation, POD is greater than 0.9, and FAR is approximately 0.5, and the rain-gauge-based models (F4-RG and H8-RG) can estimate surface precipitation well. For hourly precipitation, the R values between F4-RG and H8-RG and the ground rain-gauge observations are greater than 0.7, and the RMSE is

approximately 2.5 mm/h. For daily precipitation, the R values between F4-RG and H8-RG and the ground rain-gauge observations are approximately 0.9, and the RMSE is approximately 25 mm/d. In contrast, while the two IMERG-based models achieve good success in model development, when the surface precipitation integration results of the two models are compared with the ground rain-gauge observations, the comparison results are obviously worse than those of the rain-gauge-based models, which indicates that the ground rain-gauge data are more suitable as the model development ground truth for

the typhoon surface precipitation integration algorithm. The great performance of our typhoon-only model indicate that the surface precipitation estimation could be further improved by developing and using different models for different precipitation types.

Note that the input variables of the surface precipitation integration models include satellite observations, geographic locations and channel combinations. The key point of establishing surface precipitation integration models is how to

accurately discern the nonlinear relationship between the model input variables and precipitation. According to the importance of the variables shown by the RF algorithm used in the model development, TCRW, TP and geographic location rank much higher in importance, which is useful for confirming the accuracy of the surface precipitation integration results for typhoons. Furthermore, considering the complexity and variability of typhoon precipitating cloud systems, the vertical factors of clouds derived by passive microwave sensors (e.g., cloud water vapor profiles, cloud thickness) with

environmental conditions (e.g., wind shear, relative humidity) from a global forecast system should be introduced into the RF model to improve the QPE accuracy in the future.

## Code availability

The model in this paper is based on the random Forest data package in the Python language, and our implementation and analysis code are available upon request to the corresponding author (chao_liu@nuist.edu.cn).

## Data Availability

Fengyun-4A/AGRI data (http://satellite.nsmc.org.cn/PortalSite/Data/Satellite.aspx) and Himawari-8/AHI data (http://www.eorc.jaxa.jp/ptree/) was used to establish surface precipitation integration models. IMERG data (https://gpm.nasa.gov/data/directory) and surface precipitation observations collected from the China Meteorological Data Service Center (http://data.cma.cn/en) were used as predictands of surface precipitation integration algorithm.

## Supplement

The supplement related to this article is available online at: https://amt.copernicus.org/

## Author contributions

Conceptualization, C.L.; designed the study layout; methodology, Y.Z. and C.L. ; Data Curation: D.D. and Y.Z.; software, Y.Z.; validation, Y.Z., C.L. and Z.M.; formal analysis, Y.Z., D.D., Z.M., and S.T.; writing—original draft preparation, Y.Z.; writing—review and editing, Y.Z, C.L., D.D., Z.M. and S.T.; visualization, D.D. and Z.M.; supervision, C.L.; funding acquisition, C.L.. All authors have read and agreed to the published version of the manuscript.

## Competing interests

The authors declare that they have no conflict of interest.

## Financial support

This work was supported in part by the National Key Research and Development Program of China under Grant 2018YFC1506502, National Natural Science Foundation of China under Grants 42122038 and 41975025.

## Acknowledgements

We would like to express gratitude to NSMC, JAXA and NASA for providing AGRI, AHI and IMERG products. We also thank CMA for providing surface precipitation observation data.



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



**Table 1: Channel information for F4/AGRI and H8/AHI considered in our integration algorithms.**

| Band name | Central wavelength (μm) | |
|---|---|---|
| | F4/AGRI | H8/AHI |
| WV1 | 6.2 | 6.2 |
| WV2 | 7.1 | 7.0 |
| LW1 | 8.5 | 8.6 |
| LW2 | 10.7 | 11.2 |
| LW3 | 12.0 | 12.3 |
| LW4 | 13.5 | 13.3 |



**Table 2: Information on the three examples of typhoon events that made landfall in South China in 2018.**

| Typhoon name | Period | No rain | Rain<=5 mm/h | Rain>5 mm/h |
|---|---|---|---|---|
| Ewiniar | 6-7 Jun. 2018 (48 hrs) | 55.7% | 35.9% | 8.4% |
| Bebinca | 14-15 Aug. 2018 (48 hrs) | 78.9% | 18.3% | 2.8% |
| Mangkhut | 16 Sep. 2018 (24 hrs) | 50.9% | 36.0% | 13.1% |







**Table 3: Differences among the four precipitation integration models used in this paper.**

| Precipitation data ╲ Satellite data | FengYun-4A/AGRI (F4-based models) | Himawari-8/AHI (H8-based models) |
|---|---|---|
| Rain-gauge observations (RG-based models) | F4-RG | H8-RG |
| IMERG estimates (IMERG-based models) | F4-IM | H8-IM |


**Table 4: Data and variables considered in this study for precipitation integration.**

| | Parameters | Type | Resolution |
|---|---|---|---|
| | Longitude, Latitude | Geographic Location | - |
| Predictor | WV1, WV2, LW1, LW2, LW3, LW4, WV1-LW2, LW1-LW2, WV2-LW3, LW2-LW3 | AGRI observation | 4 km/1 h |
| | | AHI observation | 5 km/1 h |
| | CAPE, K-Index, TCRW, TP, TCLW, R850, R950, T850, T950 | Atmospheric Reanalysis | 0.25°/1 h |
| Predictand | Rain-gauge data | Rain-gauge-based | - |
| | IMERG data | IMERG-based | 0.1°/1 h |
| Validation | Rain-gauge data | Ground-based | - |




**Table 5: Surface precipitation overestimation percentages for the four models during the three typhoons.**

| Typhoon | Rain Rate | F4-RG | H8-RG | F4-IM | H8-IM |
|---|---|---|---|---|---|
| Ewiniar | < 5 mm/h | 87% | 87% | 86% | 86% |
| | > 5 mm/h | 22% | 23% | 19% | 19% |
| Bebinca | < 5 mm/h | 88% | 89% | 85% | 84% |
| | > 5 mm/h | 11% | 9.1% | 7.2% | 6.7% |
| Mangkhut | < 5 mm/h | 89% | 88% | 90% | 90% |
| | > 5 mm/h | 25% | 24% | 16% | 15% |
| Total | < 5 mm/h | 88% | 88% | 87% | 87% |
| | > 5 mm/h | 19% | 19% | 14% | 14% |


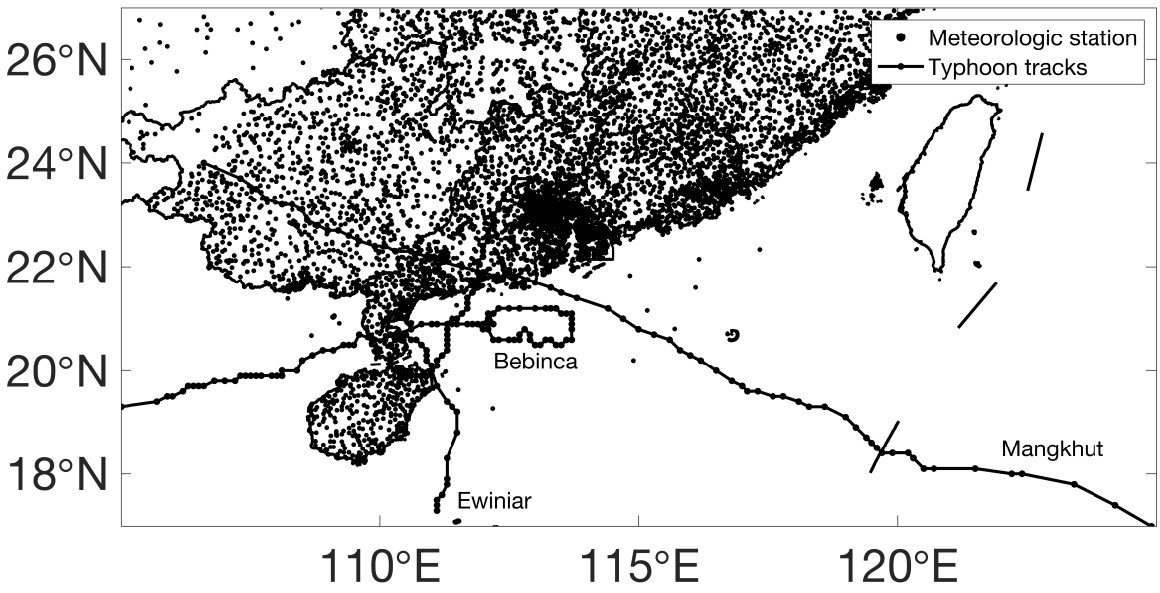

**Figure 1: Distribution of high-density automatic stations over the studied area.**







**Figure 2: Flowchart of the surface precipitation integration algorithm.**



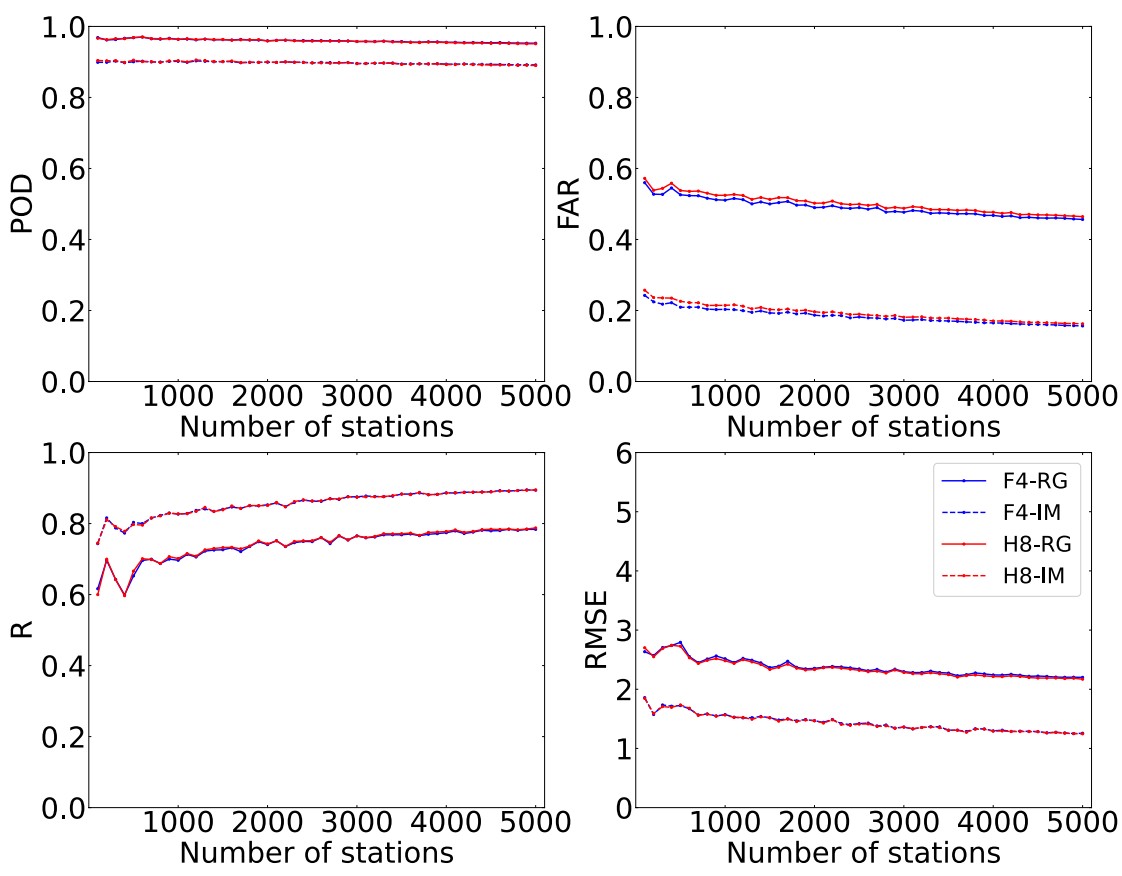


**Figure 3: Influence of the number of ground stations on the model training results.**



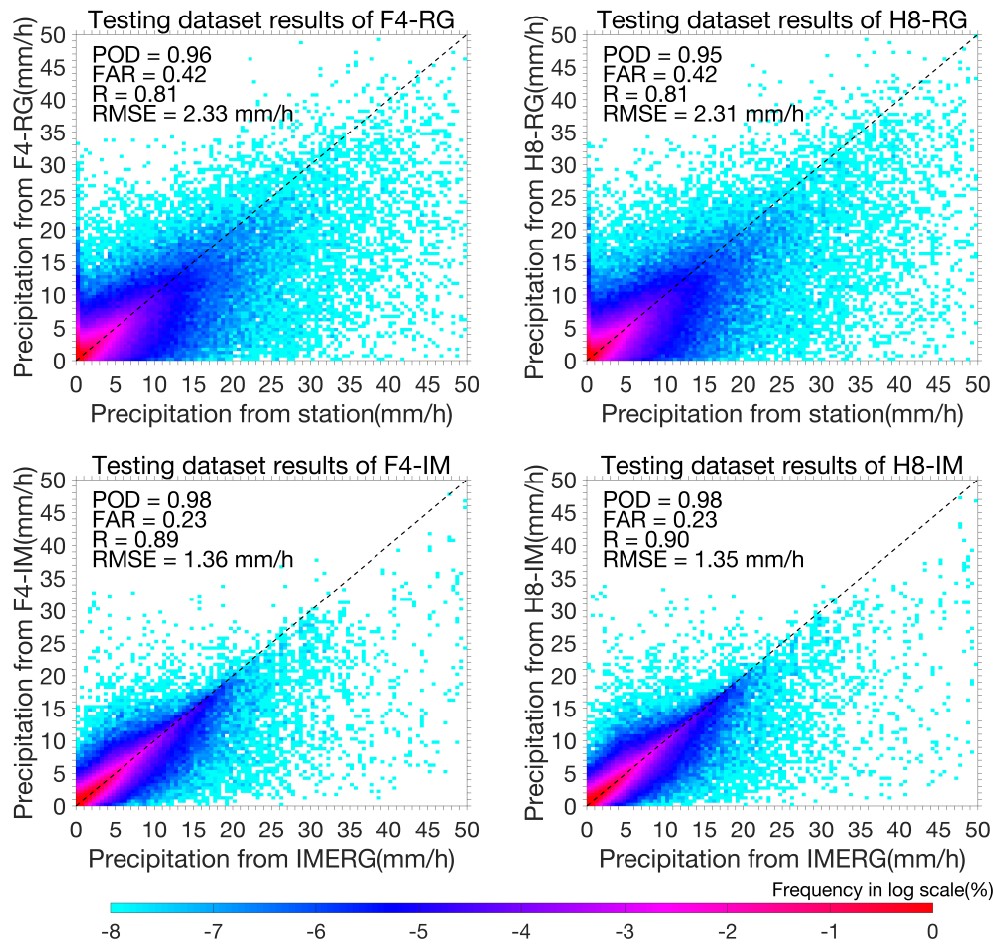

**Figure 4: Scatter plots of the hourly precipitation between different surface precipitation integration models (F4-RG, H8-RG, F4-IM and H8-IM) and rain-gauge observations over the testing datasets created by the 10-cv method during the RF algorithm. The black dotted line in all panels represents the 1:1 line.**





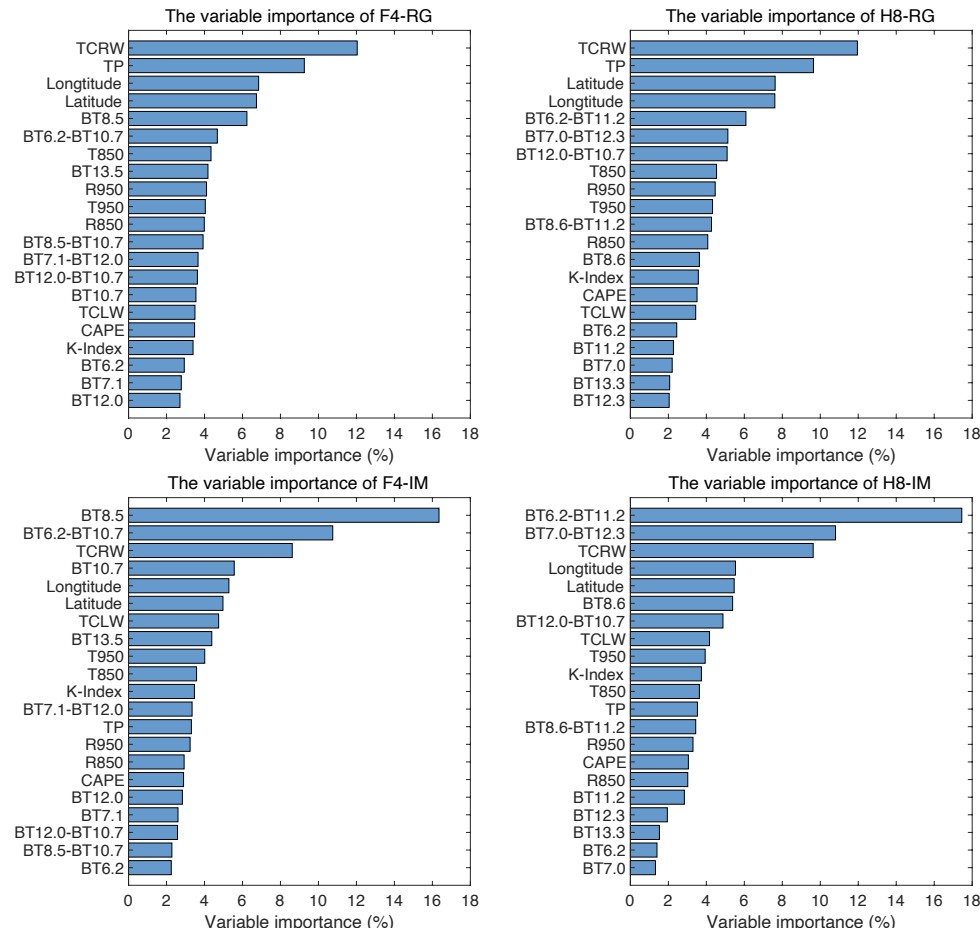

Figure 5: Importance of each variable in the RF algorithm when estimating the surface precipitation.





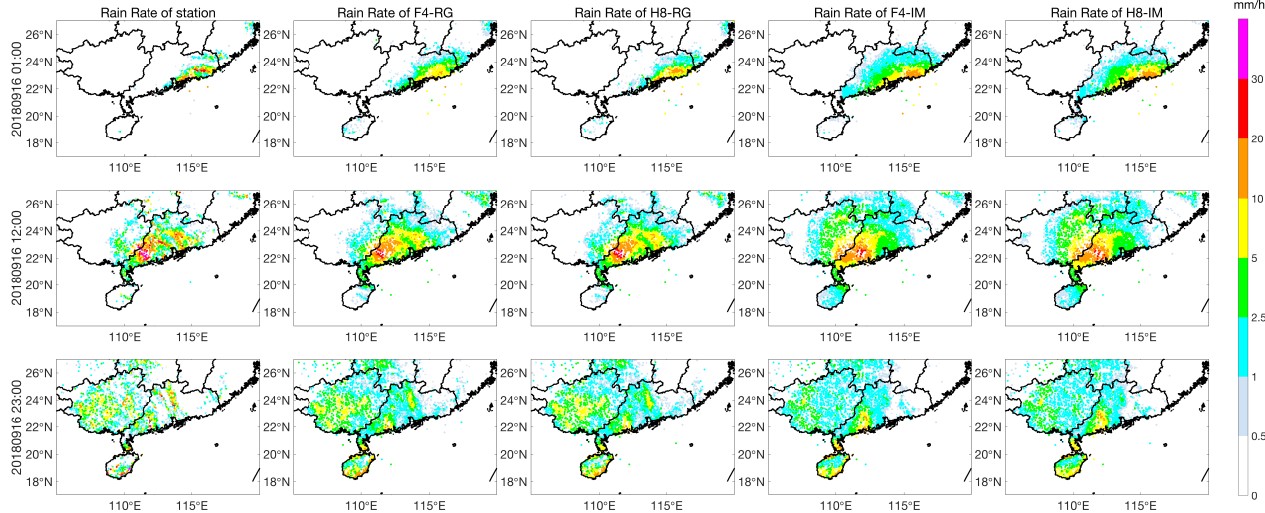

**Figure 6: Comparison of the hourly precipitation from the ground rain-gauge observations with the F4-RG, H8-RG, F4-IM and**
**H8-IM estimates at three time steps (0100 UTC on 16 Sep 2018, 1200 UTC on 16 Sep 2018, and 2300 UTC on 17 Sep 2018).**



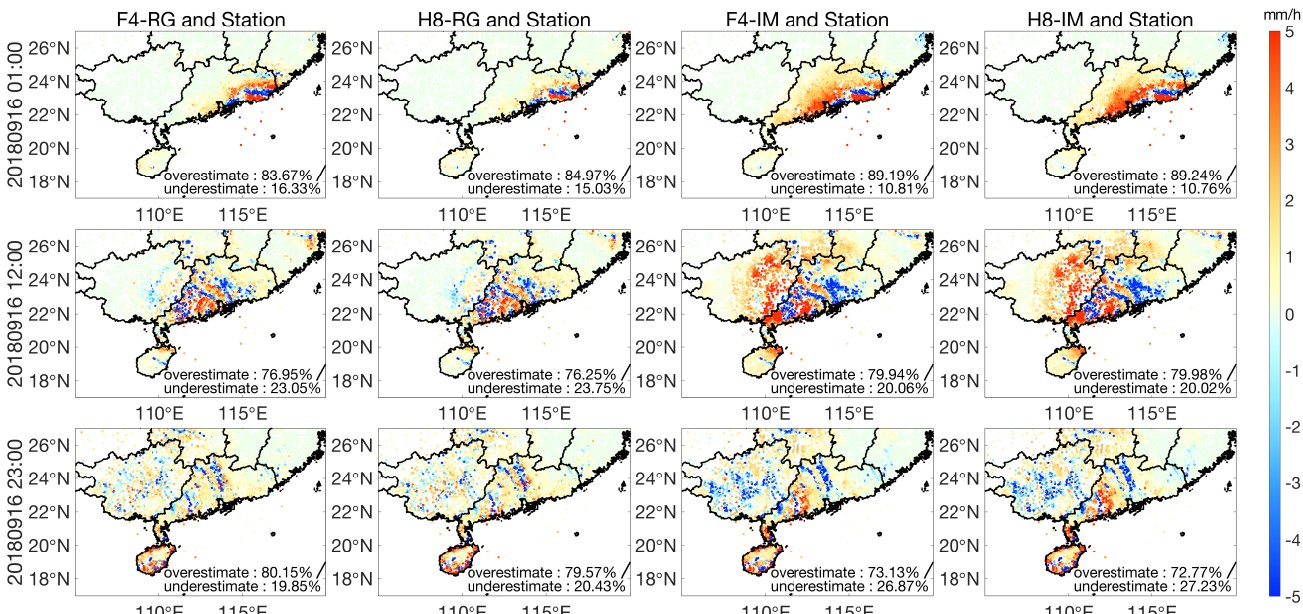

**Figure 7: Differences in hourly precipitation between F4-RG, H8-RG, F4-IM and H8-IM and the ground rain-gauge observations at three time steps (0100 UTC on 16 Sep 2018, 1200 UTC on 16 Sep 2018, and 2300 UTC on 17 Sep 2018).**





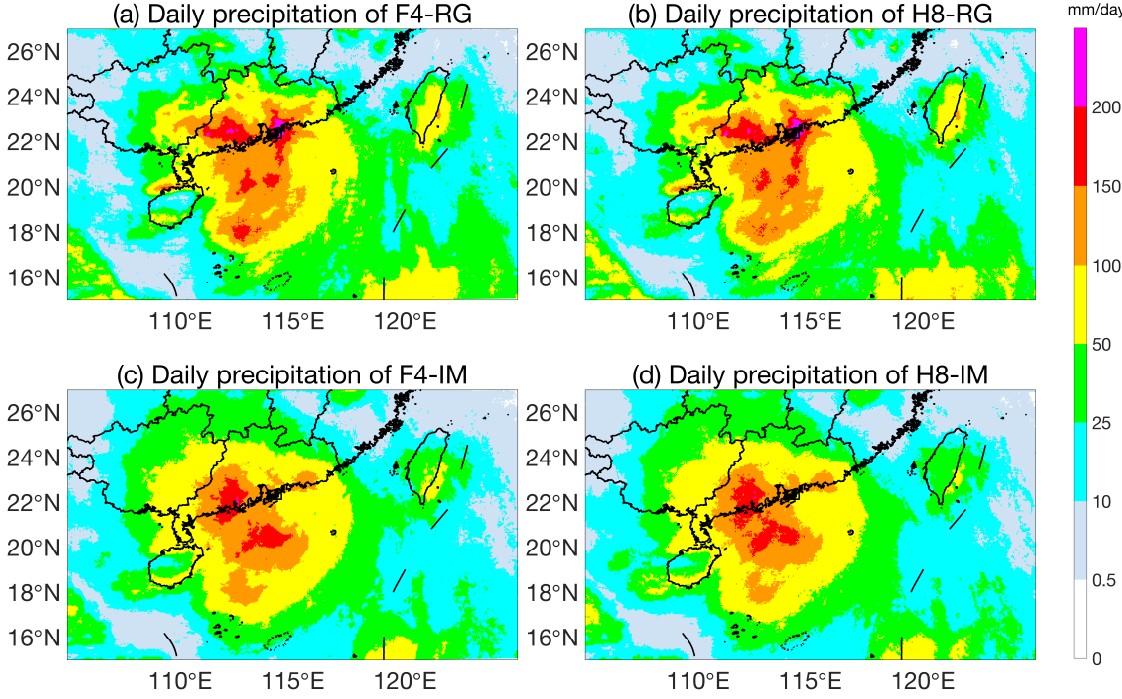

**Figure 8: Comparison of daily precipitation among F4-RG, H8-RG, F4-IM and H8-IM on 16 Sep 2018.**





**Figure 9: Differences in daily precipitation between F4-RG, H8-RG, F4-IM and H8-IM and the ground rain-gauge observations on 16 Sep 2018.**





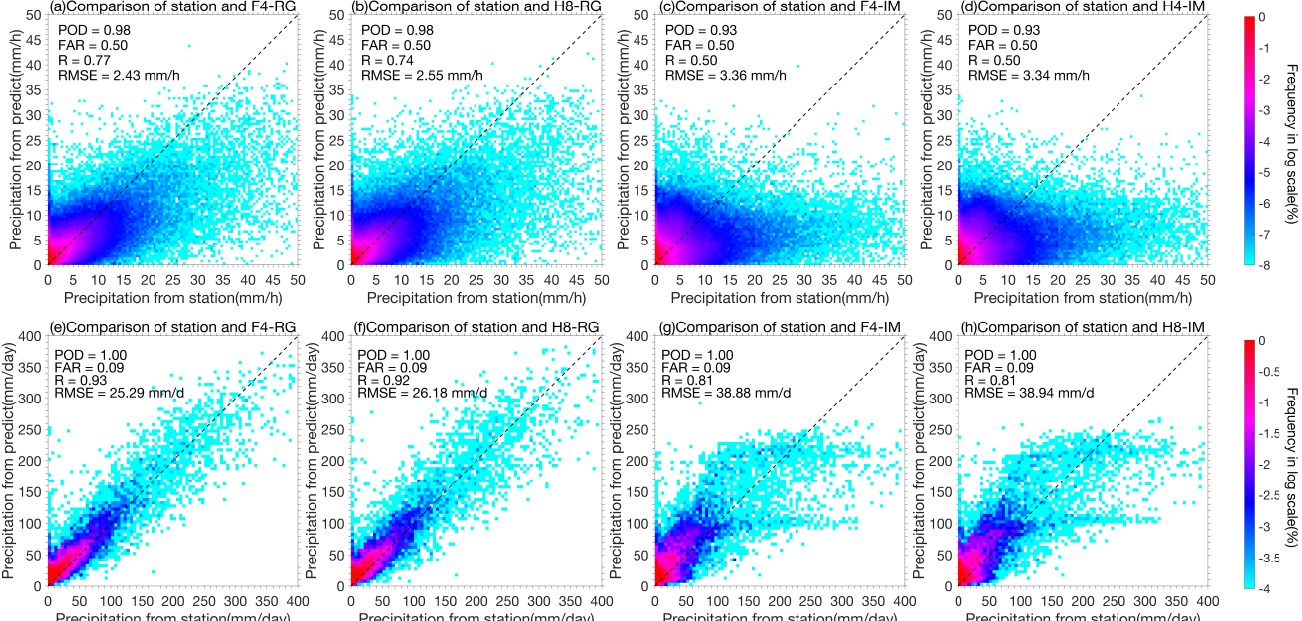

**Figure 10: Probability density distribution of the hourly ((a)~(d)) and daily ((e)~(h)) precipitation of F4-RG, H8-RG, F4-IM and H8-IM during three typhoon events. The black dotted line in all panels represents the 1:1 line.**

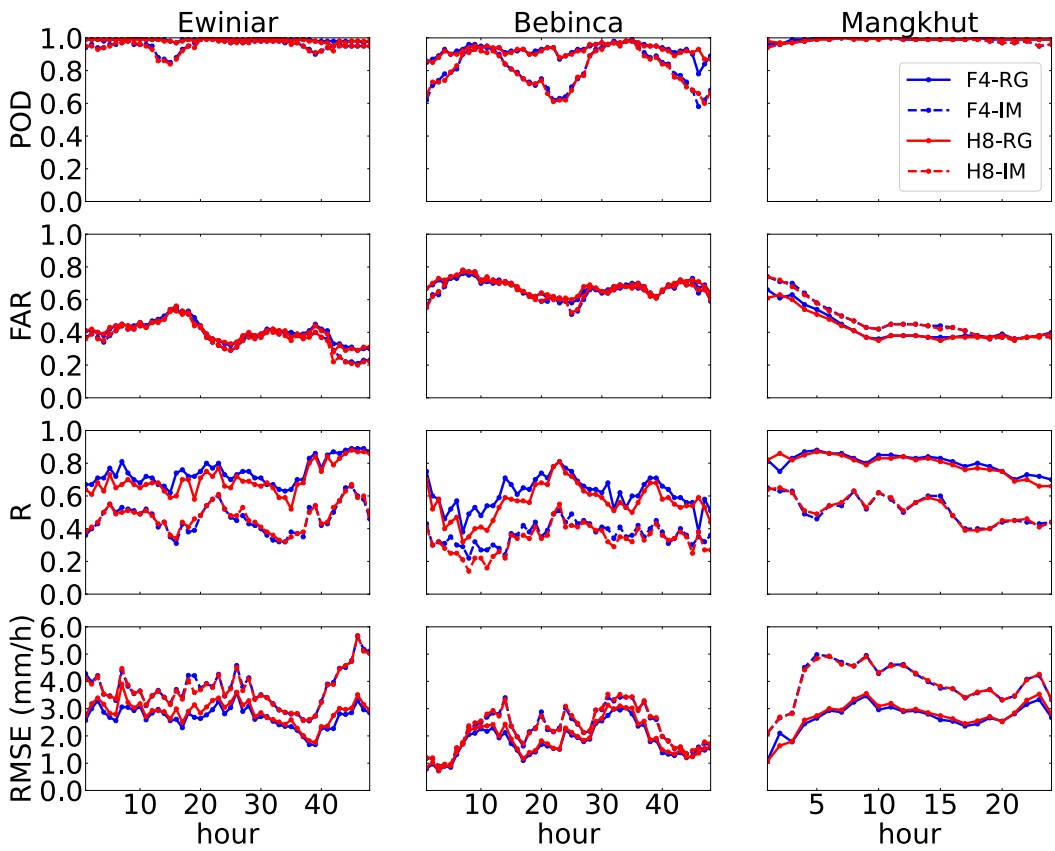

**Figure 11: Time series plots of the POD, FAR, R, and RMSE (mm/h) from F4-RG, H8-RG, F4-IM and H8-IM throughout the evolution of the three typhoon events.**
