# Peer review of "High-resolution typhoon precipitation integrations using satellite infrared observations and multi-source data"

_Atmospheric Measurement Techniques, 2021_

## Author Comment (AC1)

**Response to Reviewer #1 (amt-2021-411)**

First of all, we would like to thank the editor and reviewers for their valuable comments. We have taken all the suggested changes into consideration and revised the manuscript accordingly. The reviewers' comments are copied here as texts in BLACK, our responses are followed in BLUE, and the major corrections are marked in RED in the manuscript.

1. How much differences would different RF parameters be introduced for the model, and this should be briefly discussed.

**Response**: Thanks for the suggestion, and we have tested our models with different RF parameters, which are illustrated in Figure 3 (also see below) of the revision. As shown in the figure, the model performance becomes almost consistent after the $N_{tree}$ *and* $M_{feature}$ values becoming larger than ~200 and ~10 respectively. See the revision for details.

[Figure]

**Figure R1** (Figure 3 in the revision): Dependence of the correlation coefficient (R) on the parameters $N_{tree}$ (a) and $M_{feature}$ (b).

2. Line 221: The session title is suggested to be rephrased.

**Response**: Sorry for the confusion, and we have modified the title as "Performance for typhoon precipitation integrations".

3. As noticed in Figure 8, once the models are developed, it would also work for precipitation over ocean but maybe with less accuracy. This should be clarified in the paper.

**Response**: Thanks for the suggestion, and we have added the following discussion in the revision:

"*It is worth noting that our models can give precipitation distribution over ocean as well, while its performance could hardly be fairly evaluated due to the lack of the ground-based observations.*"

4. There are some unnecessary texts in Figure 9(a), which should be removed.

**Response**: Thanks, and we have updated the figure.

---

## Author Comment (AC2)

**Response to Reviewer #2 (amt-2021-411)**

First of all, we would like to thank the editor and reviewers for their valuable comments. We have taken all the suggested changes into consideration and revised the manuscript accordingly. The reviewers' comments are copied here as texts in BLACK, our responses are followed in BLUE, and the major corrections are marked in RED in the manuscript.

The authors compared different machine learning models to integrate data from satellites and in situ observations to estimate precipitation during typhoon landfall. The results are well presents and meaningful. Before the acceptance of this manuscript, the authors should provide more in depth discussions on the difference between QPE results estimated from different models as this is the core of this study.

Response: Thanks for the suggestions, and we have added much more discussions in the revision per your suggestion. The added discussions not only help readers to better understand our results, but also help us a lot.

Specific comments:

The language can be improved to enhance the readability. For instance:

Line 17: "… atmospheric reanalysis (ERA-5) via random forest (RF). The RF method fuses…"

Line 21: "… with correlation coeficients (R) of ~0.75 and probabilities of detection (POD) as large as of 0.98."

Line 23: "… due to the uncertainties in IMERG retrievals."

Line 24: "… the RF algorithm can well integrate information …"

**Response**: Sorry for the language problems, we have reorganized the abstract, and improved the language with the help of professional English editors. The readability has been significantly improved. The session is modified as the following:

*"The RF methods intend to integrate cloud and atmospheric features from radiometric observations and reanalysis information, and the intensity and spatial*

*distribution of precipitation can be revealed by high-density rain-gauge or IMERG data. We take three typhoons made landfall in South China in 2018 as examples. Both F4-based and H8-based results using rain-gauge data as the predictand show excellent results, and the correlation coefficients (R) and probabilities of detection (POD) reach ~0.75 and ~0.95 respectively, both higher than satellite-only results. Meanwhile, when the IMERG data are used as the predictand, the corresponding R and POD drop to ~0.5 and 0.93, respectively, due to the uncertainties related to IMERG retrievals. By carefully choosing the predictors, our RF algorithms successfully summarize the information from satellite observations, surface observations and atmospheric reanalysis, resulting in precipitation results that are highly consistent with the actual ground observations. Our proposed integration framework can reconstruct hourly surface precipitation estimation datasets at high-spatial resolution for historical typhoon studies.*"

Line 43: "the amount of rainfall reaching the ground

**Response**: The sentence has been modified as:

"*The ground-based rain-gauge is direct and accurate to measure local surface precipitation, while is limited by the station locations and coverages, especially in oceanic, mountainous and polar regions.*"

Line 60: the literature review regarding to QPE via machine learning methods is out of date, advancements in recent five past years should be reviewed and discussed.

**Response**: Thanks for the suggestion. We have updated the literature review as follows.

"*At present, machine learning (ML) methods are widely used to establish the relationship between precipitation and satellite passive spectral observations as well [Albawi et al., 2017; Sehad et al., 2017; Min et al. 2018; Ahmed et al., 2020; Wang et al., 2021; Zhang et al., 2021].*"

Data: the authors stated that they used a set of atmospheric variables from ERA-5

reanalysis as basic inputs to RF other than radiances from satellites. As we know, there is about three month time lag to produce such reanalysis data, so how to adopt the proposed method for near real time application?

**Response**: Thanks for the suggestion. In order to fully account for the influence of atmospheric information and cloud characteristics on precipitation, we use both ERA-5 data and satellite observation data to train our target model. As mentioned by the reviewer, the reanalysis data would be possibly of a few month lag. However, we think that the proposed method is possible for near real time application, because the reanalysis data may be replaced by weather forecasting data, which could be on time for such prediction (which was not detailed in the abstract). Meanwhile, we do agree with the reviewer that such statement is less precise without the details, so we remove the discussions related to near real time estimation to avoid unnecessary confusion.

Another issue is related to the generation of atmospheric reanalysis. Satellite and ground observations are often assimilated when generating reanalysis fields, so the QPE learning model could be not properly trained when variables such as total precipitation was used.

**Response**: We intend to use the ML method to integrate multi-source data, and to obtain precipitation products with high precision and spatial resolution. The reanalysis assimilates observations and model simulations to provide systematic atmospheric variables and data, and, with a significantly amount of multi-source observations, we think the reanalysis data become almost independent for AHI and AGRI spectral observations. However, those variables from reanalysis are only a part of our inputs, and we take the advantages of the ML methods to integrate such complex information. However, the reviewer raises a really interesting point, and we would definitely investigate this in the future.

Line 170: references are needed to support such claims "… but have little influence on the prediction results", "Changes in these two parameters have little influence on our modeling results, so we will not discuss the sensitivities of results to the RF

parameters". Actually, the model structure plays crucial roles in determining the final learning accuracy, so the statements given above are a bit arbitrary.

**Response**: Thanks for the suggestion. Sorry for the confusion, we made extensive test runs during model development, while didn't include the details in the manuscript. Per the reviewers' suggestion, we added the results for parameter tests in the revision. As seen in the new Figure 3 as well as the discussions, the above statements are well supported.

Figure 4: as shown, the model with rain gauge measurements used as the predictand had an accuracy lower than that of IMERG, commonly we treat in situ measurements as the ground truth and thus the accuracy from the former should have a nominal higher accuracy. Is this due to spatial inconsistency between gridded data and rain gauge data.

**Response:** There are accuracies of two kinds that briefly represent model performance: one for the model development (left part of Figure 2, i.e., the accuracy mentioned here by the reviewer) and one for the model estimation (right part of Figure 2). The former is mostly dependent on the correlation between the predictors and predictands, which should be better referred to as the model's generalization ability, while later depends on not only the predictors and predictands correlations, but also the accuracy of predictands themselves with respect to the truth. The "better performance" here only illustrates the better correlations between the predictors and predictants, but is not a high accuracy of the final model estimations, which also depend on the accuracy of the predictants compared to the truth (see Section 4). **In other words**, the IM-based methods have higher accuracy during the model development, because the IMERG results (predictand values) are highly correlated to the predictors. As expected also by the reviewer, the RG-based models are more accurate for model estimation, because their predictand values are more accurate compared to the truth. We have clarified those in the revision as the following.

Line 227–228 and Figure 5: what are reasons behind different dominant variables in

four learning models? Needs to discuss in depth.

**Response**: Thanks for the suggestion. The importance/dominance here indicates the correlation between a certain predictor and the predicant, so the most important reason for the different dominant variables is the different predictand values used between RG- and IM-based models. Because IM-based predicant values are from satellite observations, so they better correlate to satellite spectral observations and distribution, corresponding to higher rankings of satellite observations. Meanwhile, the higher rankings of reanalysis-based variables in the RG-based models indicate their reasonable performance for distributing accurate atmospheric states (i.e., surface precipitation from rain gauges). In the revision, we added some discussions as the following:

*"Since the IMERG data is a satellite-based precipitation estimation product (Min et al., 2018), there is a high consistency between IMERG data and satellite observation, which is also reflected in the importance of variables. For F4-IM and H8-IM, the importance ranking of variables such as BT8.5, BT6.2-BT10.7, BT6.2-BT11.2 and BT7.0-BT12.3 increased significantly, while the importance ranking of atmospheric reanalysis data and geographic data decreased, resulting in the precipitation predicted by the two models being more dependent on satellite observation. Meanwhile, the higher rankings of reanalysis-based variables in the RG-based models indicate their reasonable performance for distributing accurate atmospheric states (i.e., surface precipitation from rain gauges)."*

Figures 6–9: the authors should provide more discussions on the different spatial distribution of QPE results from different models since same predictors were used.

**Response**: Thanks for the suggestion. First, the differences in the spatial distribution of precipitation in Figures 8-10 are essentially caused by the differences in Figure 7, so we only detail such differences in the discussions related to Figure 7. The differences can be understood as the following:

(1) First, the differences are introduced by the different predicants considered. As expected, the precipitation predicted by the RG-based models is closer to the

rain-gauge precipitation and the IM-based model results are closer to the IMERG precipitation.

(2) The differences between the two RG-based models are less significant, similarly for the two IM-based model results, because the F4 and H8 provide very similar satellite observations for the estimations. The discussion in the paper is as follows.

(3) The detailed differences among the four models are also described in the manuscript.

"*First, the differences are introduced by the different predictands considered, which are clearly shown between the upper and lower panels. As expected, the precipitation predicted by the RG-based models is closer to the rain-gauge precipitation and the IM-based model results are closer to the IMERG precipitation. Meanwhile, the differences between the two RG-based models (the upper panels) are less significant, similarly for the two IM-based model results (the lower panels), because the F4 and H8 provide very similar satellite observations for the estimations.*"